# Pretraining Vision-Language Model for Difference Visual Question Answering in Longitudinal Chest X-rays

**Yeongjae Cho**[*†1]                                                    YJCHO@BDAI.SNU.AC.KR
**Taehee Kim**[*2]                                                   TAEHEE@RADISENTECH.COM
**Heejun Shin**[2]                                                SHJ4901@RADISENTECH.COM
**Sungzoon Cho**[‡1]                                                       ZOON@SNU.AC.KR
**Dongmyung Shin**[‡2]                                          SHINSAE11@RADISENTECH.COM
[1] *Seoul National University*

[2] *Artificial Intelligence Engineering Division, Radisen Co. Ltd.*

**Editors:** Accepted for publication at MIDL 2024

## Abstract

Difference visual question answering (diff-VQA) is a challenging task that requires answering complex questions based on differences between a pair of images. This task is particularly important in reading chest X-ray images because radiologists often compare multiple images of the same patient taken at different times to track disease progression and changes in its severity in their clinical practice. However, previous works focused on designing specific network architectures for the diff-VQA task, missing opportunities to enhance the model's performance using a pretrained vision-language model (VLM). Here, we introduce a novel VLM called PLURAL, which is pretrained on natural and longitudinal chest X-ray data for the diff-VQA task. The model is developed using a step-by-step approach, starting with being pretrained on natural images and texts, followed by being trained using longitudinal chest X-ray data. The longitudinal data consist of pairs of X-ray images, along with question-answer sets and radiologist's reports that describe the changes in lung abnormalities and diseases over time. Our experimental results show that the PLURAL model outperforms state-of-the-art methods not only in diff-VQA for longitudinal X-rays but also in conventional VQA for a single X-ray image. Through extensive experiments, we demonstrate the effectiveness of the proposed VLM architecture and pretraining method in improving the model's performance. Our code is available at:
https://github.com/yjch00/PLURAL

**Keywords:** Vision-Language Model, Visual Question Answering, Chest X-ray, Longitudinal Data, Pretraining

## 1. Introduction

Assessing longitudinal medical images is an essential daily task for radiologists. It involves reading and comparing multiple images of the same patient taken at different times. The purpose of this assessment is to track disease progression and changes in its severity. For example, radiologists review longitudinal CT images to determine whether a patient has any malignant nodules with increased sizes, which is a common indication of lung cancer

---

[*] Contributed equally

[†] Work done while interning at Radisen Co. Ltd.

[‡] Corresponding author

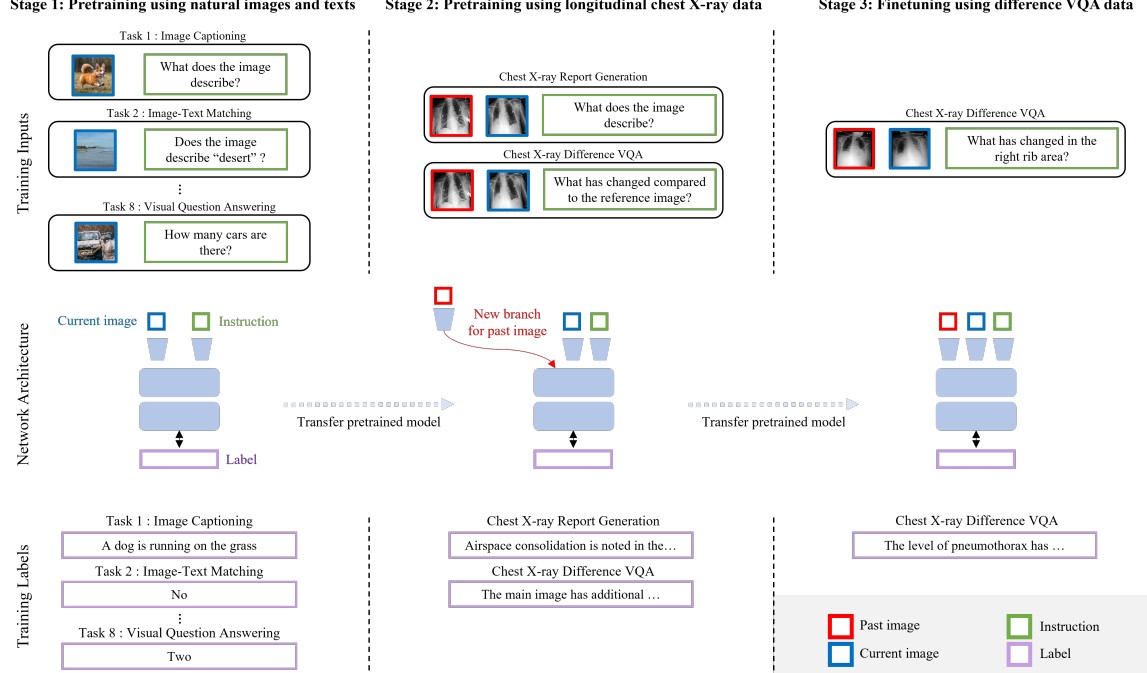

Figure 1: The training process of the PLURAL model. We adopted a Transformer-based network that takes a single image and an instruction as an input (see blue and green boxes) in the first stage. In the second and third stages, we added a new input branch for a past image (see red boxes) to utilize longitudinal chest X-ray data, including chest X-ray reports and difference VQA data.

(Callister et al., 2015). In the case of tuberculosis, they regularly compare longitudinal chest X-ray images to monitor the effect of treatment and check for any improvement in pulmonary findings (Lee et al., 2020).

Visual question answering (VQA) is a challenging task that involves answering a set of complex questions about an image. With the increasing popularity of vision-language models (VLMs) (Alayrac et al., 2022; OpenAI, 2023; Liu et al., 2023), VQA is gaining attention as an effective method to interpret medical images (Lin et al., 2023), assist visually impaired individuals (Bigham et al., 2010), and so on. In particular, a recent study (Hu et al., 2023) has attempted to solve a VQA task based on longitudinal images, detecting the changes between two X-ray images taken at different time points. This task, called difference VQA (diff-VQA), is especially helpful in assisting radiologists in their clinical practice.

In previous works (Qiu et al., 2021; Yao et al., 2022; Hu et al., 2023), specific network architectures were designed to solve the diff-VQA task effectively because this task requires an AI model to differentiate between common and distinguished features in two images. However, customizing neural networks has caused them to miss valuable opportunities to enhance the model's performance by utilizing VLMs pretrained on a vast number of texts and images, in general, (Chen et al., 2022; Wang et al., 2022; Bao et al., 2022) and/or medical domains (Zhang et al., 2023; Moor et al., 2023; Li et al., 2023). As a result, no

studies have proposed an effective pretraining pipeline for diff-VQA or investigated the impact of pretraining settings, such as pretraining steps and data composition.

In this study, we introduce a novel VLM called PLURAL, which stands for pretrained vision-language model based on natural and longitudinal chest X-ray data for diff-VQA. The training process of this model involves using a Transformer-based network (Wang et al., 2022) that has been pretrained on natural images and texts (see Stage 1 in Figure 1) and then optimizing it using longitudinal chest X-ray data (see Stage 2 and 3 in Figure 1). The longitudinal data consists of pairs of longitudinal X-rays, QA sets, as well as X-ray reports that provide detailed descriptions of the changes in lung abnormalities and diseases over time. This step-by-step training approach allows the PLURAL model to benefit from the transferred knowledge of a pretrained VLM, while also taking advantage of the rich temporal information in the X-ray reports.

We have demonstrated the effectiveness of the PLURAL method by comparing its performance in the diff-VQA data with that of previous state-of-the-art (SOTA) methods. We conducted several ablation studies to confirm the impact of each pretraining component, such as natural data pretraining, longitudinal data pretraining, inputs of past images, and some sections in the X-ray reports (i.e., *Findings* and *Impression*). Through extensive experiments, we showed that the PLURAL model outperformed the other SOTA methods not only in diff-VQA for longitudinal X-rays but also in conventional VQA for a single X-ray image.

## 2. Method

The training process of the PLURAL model includes a series of three training stages as follows (see Figure 1): pretraining using natural images and texts, pretraining using longitudinal chest X-ray data, and finetuning using diff-VQA data. To train the network using single or longitudinal images according to the different stages, we adopted a Transformer-based network (Wang et al., 2022) in the first stage and modified its inputs in the second and third stages (see Stage 2 and 3 in Figure 1).

### 2.1. Vision-Language Model Architecture

The VLM architecture we used in the second and third stages is shown in Figure 2. Based on a Transformer encoder-decoder architecture proposed by Wang et al. (2022), we designed our own architecture by attaching a new input branch for a past image (i.e., input branch for $i_{past}$ in Figure 2). This design allowed the model to take two longitudinal images as input simultaneously.

Both past ($i_{past} \in R^{H \times W}$ where $H = 384$ and $W = 384$) and current ($i_{cur} \in R^{H \times W}$) images are encoded using the same image encoder ($E_{img}(i_{past}) \in R^{\hat{H} \times \hat{W} \times E}$ where $\hat{H} = 24$, $\hat{W} = 24$, and $E = 1024$), which is a ResNet101(He et al., 2016), and were flattened ($v_{past} = F(E_{img}(i_{past})) \in R^{N \times E}$ where $N = \hat{H} \cdot \hat{W} = 576$). Then, the flattened feature vector for each image is linearly projected ($P_{img}(v_{past}) \in R^{N \times D}$ where $D = 768$), followed by the addition of the positional ($p_{enc}^{img} \in R^{N \times D}$) encoding. After that, to differentiate the time points of the two input images, we separately added the time encoding for each image

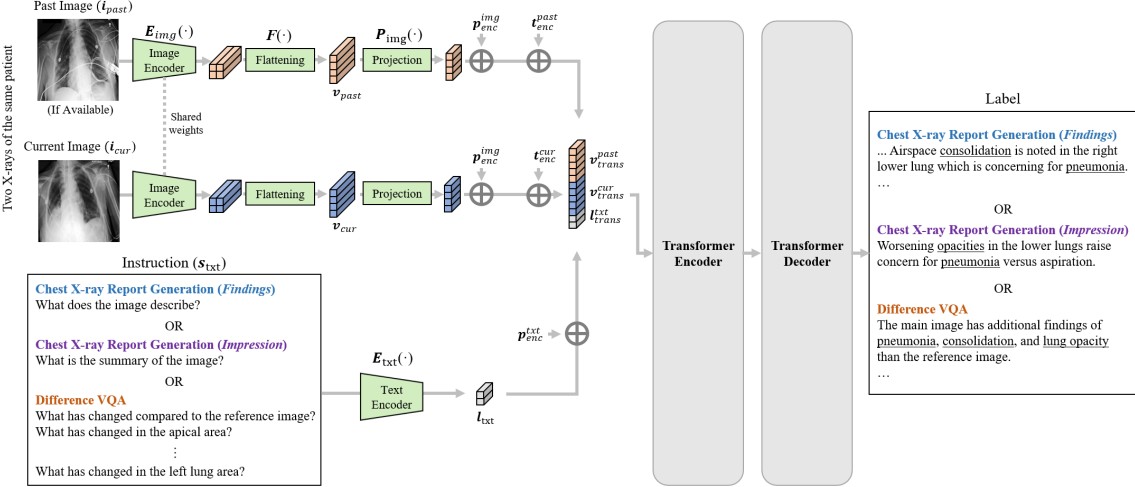

Figure 2: Proposed VLM architecture used in the second and third training stages of PLURAL. We modified a basic Transformer architecture by adding a new input branch for a past image. This enabled the model to take concurrently two longitudinal chest X-ray images as an input.

$(t_{enc}^{past} \in R^{N \times D}$ for past; $t_{enc}^{cur}$ for current) as trainable parameters:

$$v_{trans}^{past} = P_{img}(v_{past}) + p_{enc}^{img} + t_{enc}^{past} \tag{1}$$

$$v_{trans}^{cur} = P_{img}(v_{cur}) + p_{enc}^{img} + t_{enc}^{cur} \tag{2}$$

The text instruction ($s_{txt}$) was encoded using the text encoder ($l_{txt} = E_{txt}(s_{txt}) \in R^{N_t \times D}$ where $N_t$ is the length of an instruction sequence; byte-pair encoding (Sennrich et al., 2015) and word embedding used), followed by the addition of the positional encoding ($p_{enc}^{txt} \in R^{N_t \times D}$, $l_{trans}^{txt} = l_{txt} + p_{enc}^{txt}$). Finally, all the processed features of the input images and instruction were aggregated, producing the concatenated input feature ($i_{trans} \in R^{(2 \cdot N + N_t) \times D}$) for the Transformer encoder:

$$i_{trans} = concat(v_{trans}^{past}, v_{trans}^{cur}, l_{trans}^{txt}) \tag{3}$$

The Transformer encoder (6 layers) subsequently encoded the combined feature ($i_{tras}$), and the Transformer decoder (6 layers) disentangled the output of the Transformer encoder to generate an answer to the instruction. The loss function was a cross-entropy loss as follows: $\mathcal{L} = -\sum_{j=1}^{|y|} \log P_\theta(y_j | y_{<j}, i_{past}, i_{cur}, s_{txt})$, where $\mathcal{L}$ is a cross-entropy loss, $y$ is an output of the Transformer decoder, and $\theta$ is trainable model parameters.

## 2.2. Stage 1: pretraining using natural images and texts

In the first stage of the training process of PLURAL, the network was pretrained using various tasks (eight tasks including image captioning, image-text matching, VQA, etc.) and datasets of natural images and texts (e.g., CC12M dataset (Changpinyo et al., 2021), SBU dataset (Ordonez et al., 2011), COCO dataset (Lin et al., 2014), etc.) as described in the work of Wang et al. (2022). Since all the tasks in the first stage consider a single

image (or no image) as an input, in this stage, the network architecture included neither an input branch nor time encoding for a past image (i.e., no input branch for $i_{past}$ in Figure 2). The loss function was defined only to consider a single input image as follows: $\mathcal{L} = -\sum_{j=1}^{|y|} \log P_\theta(y_j|y_{<j}, i_{cur}, s_{txt})$, which is different from the loss function used in the second and third stages. In our implementation, we utilized a pretrained model weight available on the web[1] (OFA$_{base}$ model with 184M parameters) instead of training the model from scratch by ourselves. Detailed training parameters (e.g., a learning rate, an optimizer, and a dropout ratio) can be found in Wang et al. (2022).

### 2.3. Stage 2: pretraining using longitudinal chest X-ray data

In the second stage, we continued to pretrain the model from the first stage using longitudinal chest X-ray data. To do that, we added a new input branch for a past image (see Stage 2 in Figure 1), modifying the network architecture as shown in Figure 2.

We utilized two sets of longitudinal data for pretraining, consisting of longitudinal chest X-rays with QA pairs (MIMIC-Diff-VQA dataset (Hu et al., 2023); see Section 2.5 for details) and radiologist's reports (MIMIC-CXR dataset (Johnson et al., 2019)). Two main sections, *Findings* and *Impression*, in the chest X-ray reports were used for pretraining. We introduced separate instructions for each section, "What does the image describe?" for the *Findings* section and "What is the summary of the image?" for the *Impression* section. These sections include valuable information about the longitudinal changes in chest X-ray images, which can serve as important clues to solve the diff-VQA task effectively. For example, as shown in the 'Label' box in Figure 2, they imply many indications for the additional findings, such as consolidation, opacity, and pneumonia (see underlying words in Figure 2) that are the key answers to the difference VQA question (e.g., "What has changed compared to the reference image?").

### 2.4. Stage 3: finetuning using difference VQA data

In the last training stage of PLURAL, the model from the second stage was finetuned using only the diff-VQA data without any modification of the network architecture (see Stage 3 in Figure 1). Details of training configurations are summarized in Appendix C.

### 2.5. Longitudinal Chest X-ray Datasets

**MIMIC-CXR** (Johnson et al., 2019) is a large-scale chest X-ray dataset comprising 377,110 chest X-ray images and 227,827 reports from 63,478 patients. Each report contains two main sections: *Findings*, which contains a detailed description of the image, and *Impression*, which provides a concise summary of the findings. We utilized the training and validation data based on the official split to train the PLURAL model in the second stage. We extracted the *Findings* sections from the reports and performed preprocessing (e.g., eliminating special characters, typos, and scarce words), following the work of Chen et al. (2020). To extract the *Impression* sections, we followed the method of Endo et al. (2021). We only used chest X-ray images of posterior-anterior or anterior-posterior views, discarding those of other views such as lateral. For each pair of an X-ray image and a report

---

1. https://github.com/OFA-Sys/OFA

Table 1: Comparison of AI performance between previous SOTA methods and PLURAL in diff-VQA.

| Model | BLEU-1 | BLEU-2 | BLEU-3 | BLEU-4 | METEOR | ROUGE-L | CIDEr |
|---|---|---|---|---|---|---|---|
| MCCFormers (2021) | 0.214 | 0.190 | 0.170 | 0.153 | 0.319 | 0.340 | 0 |
| IDCPCL (2022) | 0.614 | 0.541 | 0.474 | 0.414 | 0.303 | 0.582 | 0.703 |
| EKAID (2023) | 0.628 | 0.553 | 0.491 | 0.434 | 0.339 | 0.577 | 1.027 |
| PLURAL | **0.704** | **0.633** | **0.575** | **0.520** | **0.381** | **0.653** | **1.832** |

Table 2: Results of the ablation study when we removed each pretraining stage in PLURAL.

| Model | Pretraining | | BLEU-1 | BLEU-2 | BLEU-3 | BLEU-4 | METEOR | ROUGE-L | CIDEr |
| | Natural images and texts | Longitudinal chest X-ray data | | | | | | | |
|---|---|---|---|---|---|---|---|---|---|
| (a) | | | 0.651 | 0.575 | 0.513 | 0.455 | 0.341 | 0.588 | 1.088 |
| (b) | ✓ | | 0.682 | 0.611 | 0.552 | 0.496 | **0.381** | 0.640 | 1.733 |
| (c) | | ✓ | 0.680 | 0.612 | 0.554 | 0.501 | 0.370 | 0.647 | 1.796 |
| PLURAL | ✓ | ✓ | **0.704** | **0.633** | **0.575** | **0.520** | **0.381** | **0.653** | **1.832** |

(i.e., *Findings* or *Impression*), a chest X-ray of the same patient taken at a prior visit was selected and allocated to construct a longitudinal image dataset. We also utilized a single X-ray image to pretrain the PLURAL model in cases where there were no prior visits. We excluded all the images (and corresponding reports) included in the test sets of the VQA dataset (i.e., MIMIC-Diff-VQA). In other words, the final test set for diff-VQA was totally independent of this dataset.

**MIMIC-Diff-VQA** (Hu et al., 2023) is a VQA dataset containing 700,703 QA pairs about 164,324 sets of longitudinal chest X-ray images. The images are a subset of those of the MIMIC-CXR dataset. QAs were created using an automatic algorithm based on radiologist's reports. Those QA pairs are categorized as difference, presence, abnormality, view, location, level, and type based on the forms of questions. We used 131,563 and 16,372 QA pairs of 'difference' form to train and validate the PLURAL model during the second and third stages (see Figure 1) and 16,389 pairs as an independent test set to measure the performance of the model, based on the official split of the dataset.

## 3. Experiments

### 3.1. Difference Visual Question Answering

As shown in Table 1, PLURAL achieved the highest scores in all the natural language generation (NLG) metrics, including BLEU (Papineni et al., 2002), METEOR (Banerjee and Lavie, 2005), ROUGE-L (Lin, 2004), and CIDEr (Vedantam et al., 2015), compared to previous state-of-the-arts methods, such as MCCFormers (Qiu et al., 2021), IDCPCL (Yao et al., 2022), and EKAID (Hu et al., 2023). The differences between the scores were significant (e.g., CIDEr: 1.832 for PLURAL vs. 1.027 for EKAID). Additionally, Figure 3 showcases selected examples to qualitatively compare the AI outputs of EKAID and PLURAL. We discovered that, on average, PLURAL was better at capturing the longitudinal change of multiple lung findings (see Figure 3(a) and Figure 3(b)) and the severity level of abnormalities (see Figure 3(c)).

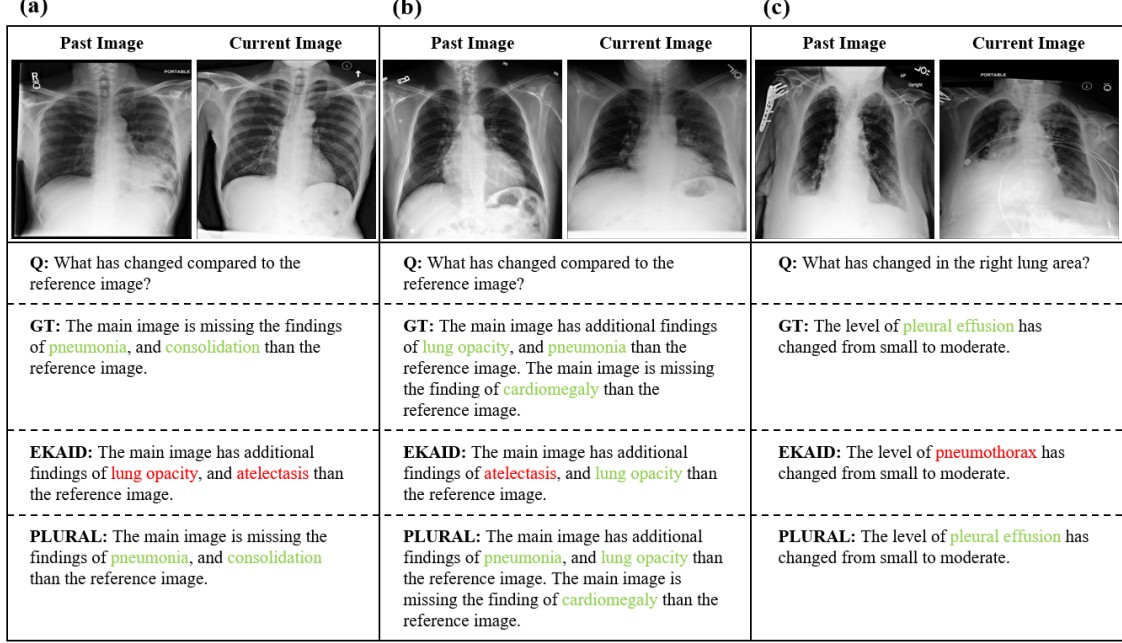

Figure 3: Selected test cases to qualitatively compare the outputs of EKAID and PLURAL in diff-VQA. We found that PLURAL was better at capturing the longitudinal change of multiple lung findings (a), (b), and change of the severity level of abnormalities, such as worsening of pleural effusion in the right low part of the lung (c).

Table 3: Results of the ablation experiments to investigate how the structure of inputs in the pretraining step (Stage 2 in Figure 1) affects the PLURAL model's performance.

| Model | Past Image | Chest X-ray Report Findings | Chest X-ray Report Impression | BLEU-1 | BLEU-2 | BLEU-3 | BLEU-4 | METEOR | ROUGE-L | CIDEr |
|-------|-----------|---------|------------|--------|--------|--------|--------|--------|---------|-------|
| (a) | | | | 0.682 | 0.611 | 0.552 | 0.496 | 0.381 | 0.640 | 1.733 |
| (b) | | ✓ | ✓ | 0.695 | 0.623 | 0.562 | 0.504 | 0.370 | 0.635 | 1.728 |
| (c) | ✓ | | ✓ | 0.678 | 0.611 | 0.557 | 0.505 | **0.391** | 0.650 | 1.731 |
| (d) | ✓ | ✓ | | 0.696 | 0.627 | 0.571 | 0.517 | 0.388 | **0.656** | **1.847** |
| PLURAL | ✓ | ✓ | ✓ | **0.704** | **0.633** | **0.575** | **0.520** | 0.381 | 0.653 | 1.832 |

## 3.2. Ablation Study

We performed ablation studies to assess the impact of pretraining steps in PLURAL. We removed each pretraining stage individually to check how it affected performance. When we removed all pretraining stages and only performed the finetuning stage (Stage 3 in Figure 1), the model reported the lowest performance (see (a) in Table 2). However, incorporating each pretraining step significantly improved model performance. For instance, adding the pretraining stage using longitudinal chest X-ray data led to a significant increase in performance (e.g., CIDEr: 1.088 for model (a) vs. 1.796 for model (c) in Table 2). The best performance was achieved when both pretraining stages were combined (see the PLURAL model in Table 2).

We also conducted ablation experiments to investigate how the structure of inputs in the pretraining stage affects the PLURAL model's performance. Specifically, we removed past

Table 4: Comparison of AI performance (exact-match accuracy) between the PLURAL and two other SOTA methods (MMQ and EKAID) on non-difference VQA.

| Model | Open Question (%) | Closed Question (%) | All Question (%) |
|---|---|---|---|
| MMQ (2021) | 11.5 | 10.8 | 11.5 |
| EKAID (2023) | 26.4 | 79.9 | 52.5 |
| PLURAL *w/o Pretraining* | 46.2 | 65.0 | 55.3 |
| PLURAL | **51.2** | **87.3** | **68.8** |

images, *Findings*, or *Impression* sections of the X-ray reports during the model's pretraining in stage 2 (Figure 1). As shown in Table 3, the best performance in terms of BLEU scores was achieved when we used past images and all report sections together as inputs. We observed a significant improvement in performance compared to the baseline model (see (a) in Table 3) across all NLG metrics (e.g., BLEU-4: 0.496 for baseline vs. 0.520 for PLURAL). This indicates that temporal information in past images and reports was indeed useful in solving the diff-VQA task.

### 3.3. Non-difference Visual Question Answering

We conducted further training and the evaluation of the PLURAL model with two other SOTA methods, MMQ (Do et al., 2021) and EKAID (Hu et al., 2023), on six categories of questions other than the 'difference' category (i.e., 'location', 'views', etc.; see Section 2.5). These questions did not require the consideration of longitudinal differences between two X-ray images (i.e., non-difference VQA). The questions were of two types: open and closed. For open questions, answers were in free form, while for closed questions, answers were either 'yes' or 'no'. We calculated exact-match accuracy between the model outputs and ground-truths in both cases.

Table 3 summarizes the results of the evaluation of non-difference VQA. The PLURAL model achieved the highest accuracy compared to PLURAL without pretraining (e.g., 65.0% vs. 87.3% in closed questions), and the two SOTA methods (e.g., 26.4% for EKAID vs. 51.2% for PLURAL in open questions). These results indicate that the PLURAL model is effective not only in answering questions about the changes in longitudinal images but also in answering other types of questions, such as the ones related to the location of abnormalities or X-ray view (see Appendix B for examples).

## 4. Conclusion

In this study, we present a Transformer-based VLM called PLURAL that is pretrained on natural images and texts, and longitudinal chest X-ray data to solve the difference VQA problem. Our approach allows the PLURAL model to benefit from both pretrained VLM knowledge and temporal information in the X-ray reports that describe the changes in lung abnormalities and diseases over time. Throughout the experiments, we have demonstrated that the PLURAL method outperformed previous SOTA methods in both diff-VQA for longitudinal X-rays and conventional VQA for a single X-ray image. We believe that this new model and training pipeline would be beneficial in promoting the application of the difference VQA task to assist radiologists in reading longitudinal chest X-rays.

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

## Appendix A. Error Analysis and Radiologist Evaluation

To perform an error analysis, we randomly selected five cases where PLURAL failed to provide the exact-matched texts with ground truths. Then, we asked a radiologist (14 years of experience) to score the correctness (1-5) of the ground truths and the answers of PLURAL and explain the reasons for the scoring (see sample cases Figure 4). In three of five cases, the scores of PLURAL were the same as those of the GTs. In one case, PLURAL produced an output text that described the change of abnormality in time-reverse order. Surprisingly, in the last remaining case, the score of PLURAL was higher than that of the GT, implying the possibility that GTs might not be the optimal description for longitudinal chest X-rays.

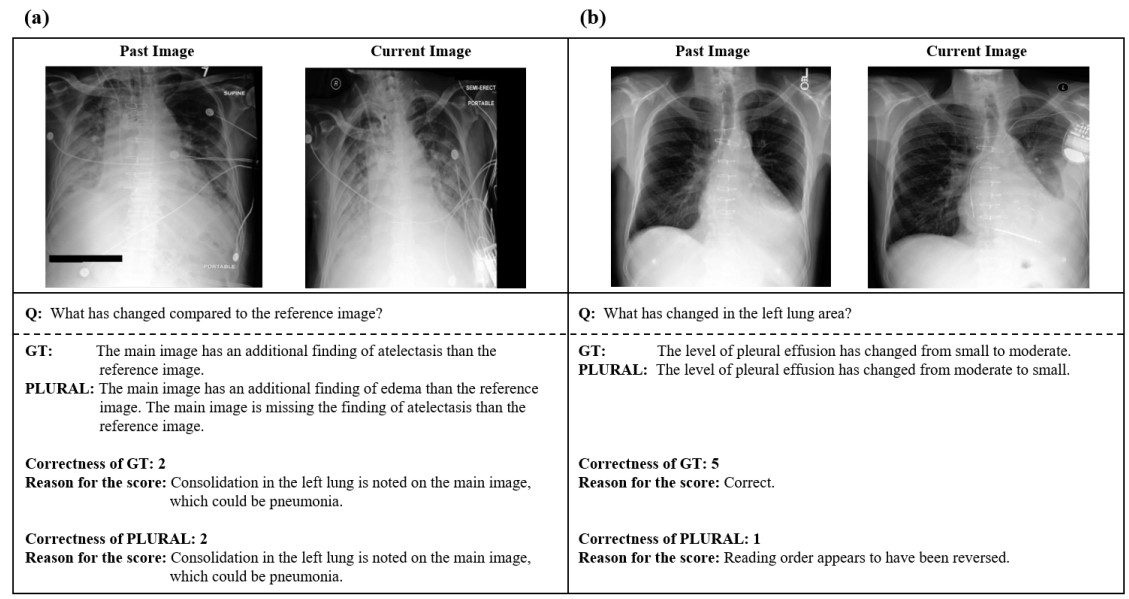

Figure 4: Radiologist evaluation of error cases: (a) GT and PLURAL with equal correctness scores; (b) Lower PLURAL score due to reversed severity order.

## Appendix B. Qualitative Analysis on Non-difference Visual Question Answering

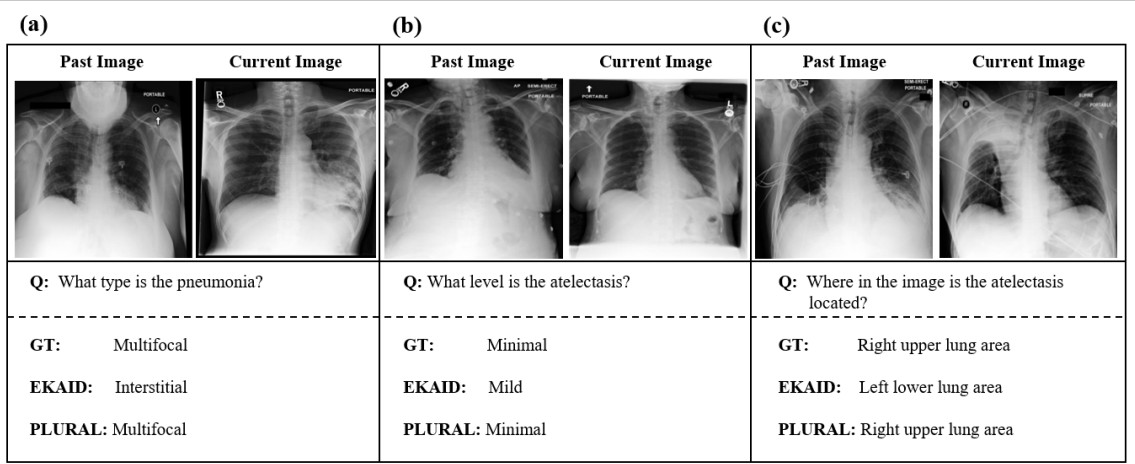

Figure 5: Selected test cases to qualitatively compare the outputs of EKAID and PLURAL in non-difference VQA. Non-difference VQA contains various categories of QA sets such as type (a), level (b), and location (c).

## Appendix C. Implementation Details

We used an AdamW (Loshchilov and Hutter, 2017) optimizer with a learning rate $= 1e-4$, $\beta_1 = 0.9$, $\beta_2 = 0.999$, and a batch size $= 16$ for pretraining and finetuning stages. The other parameters for the Transformer include a dropout ratio $= 0.1$, a weight decay $= 0.01$, and a stochastic depth $= 0.1$. In both stages 2 and 3, we stopped the training earlier when the validation loss was saturated and not improved for 5K steps. The maximum input and output text sequence length is set to 100. The hardware specifications we used were CPU = AMD Ryzen Threadripper PRO 3955WX 16-Cores, GPU = three NVIDIA A6000 GPUs with 48GB memory.

