# OpenReview forum: "Pretraining Vision-Language Model for Difference Visual Question Answering in Longitudinal Chest X-rays"
_MIDL.io/2024/Conference — MIDL 2024 Poster_

### Official Review · Reviewer_zJ5U · 2024-02-22

**Confidence:** 4
**Preliminary Rating:** 3
**Recommendation:** Poster
**Final Rating:** 3.5

**Summary:**

This paper introduces a novel VLM called PLURAL, which is pretrained on natural and longitudinal chest X-ray data for the diff-VQA task. This draft primarily builds upon EKAID to further explore pretrained Vision-Language Model (VLM) models. The motivation is sufficient, the methodology is moderately innovative, but the experimental results demonstrate state-of-the-art performance.

**Strengths:**

The paper investigates Vision-Language Model (VLM) pretrained models based on EKAID. The motivation is sound. The existing research has been thoroughly surveyed. They have demonstrated the effectiveness of the PLURAL method by comparing its performance in the diff-VQA data with that of previous state-of-the-art (SOTA) method.

**Weaknesses:**

1. Code availability is not specified.
2. It seems that the MIMIC-Diff-VQA dataset has already been used in step 2, and considering it is derived from the MIMIC-CXR dataset, it might be more beneficial for the fine-tuning in step 3.
3. The method in Figure 2 lacks sufficient innovation, as its primary framework is derived from Wang et al. 2022.

**Detailed Comments:**

1. In step 2, it appears that the MIMIC-Diff-VQA dataset has already been utilized, and considering it is derived from the MIMIC-CXR dataset, it might be more beneficial for step 3 fine-tuning. Could other datasets be considered for step 3?

2. The method in Figure 2 lacks sufficient innovation, as its primary framework is derived from Wang et al. 2022. Is there any room for partial improvements or modifications?

3. All current testing is conducted on the MIMIC dataset. Are there any other datasets available for testing?

**Justification Of Final Rating:**

Thanks for the author's efforts to reply, I am willing to revise the opinion.
The authors compare similar methods and show differences.
But I still have the problem of lack of data sets. I hope it can be reflected in my future work.

**Justification Of The Preliminary Rating:**

The criteria for evaluation stem from a comprehensive examination of the motivation and innovative aspects underlying the visual-language pretrained model. This involves a thorough exploration of the rationale behind the model's development and the unique contributions it brings to the field. Furthermore, the evaluation encompasses a comparative analysis of the model's experimental results against those of state-of-the-art methods, aiming to assess its performance and potential advancements over existing approaches.

**Questions To Address In The Rebuttal:**

See Detailed Comments. If the author can better respond to the shortcomings, I may change the grade.

**Special Issue:**

No

---

> ### Author Response · Authors · 2024-03-15
>
> Thank you for investing your time and effort in writing a truly valuable review. In this revision, we have made the code available for access (see https://github.com/yjch00/PLURAL).
>
> ## Q1 Other datasets for step 3
>
> If we understand correctly, you're asking if there are alternative datasets that could be used during the finetuning process of step 3. We had been also eager to utilize other datasets in this study; however, to the best of our knowledge, there is no other dataset that satisfies the following criteria: Longitudinal chest X-ray dataset with a particular focus on analyzing the differences between two images. A large-scale dataset that meets these specific requirements was MIMIC-Diff-VQA only. We hope to find another dataset in the near future, training and testing our model as soon as it becomes available.
>
> ## Q2 Any improvements from OFA (Wang et al. 2022) [1]
>
> We believe that the PLURAL method has the following improvement or distinct features compared to OFA: Firstly and most importantly, from the perspective of an architecture design, PLURAL has a new input branch for a past chest X-ray to capture the differences between two input images, whereas OFA doesn’t. Secondly, to process a past image more effectively, we combined a time-encoding scheme which was not used in the OFA method. Lastly, OFA adopted a simple pretraining-finetuning method to address each downstream task, while PLURAL uses a three-step pretraining pipeline which is optimized to improve the performance of AI in terms of solving diff-VQA tasks in chest X-rays.
>
> ## Q3  Other datasets available for testing
>
> As we discussed in our answer to Q1, we have really wanted to find another dataset that could be used in either the training or testing processes but failed. In line with our survey, all the previous works [2-4] utilizing longitudinal chest X-ray data to detect temporal changes have also exclusively utilized the MIMIC dataset. This is because other datasets, such as the IU Dataset [5], do not contain historical patient records, including past images, making them not practical to use in our study.
>
> [1] Wang, Peng, et al. "Ofa: Unifying architectures, tasks, and modalities through a simple sequence-to-sequence learning framework." *International Conference on Machine Learning*. PMLR, 2022.
>
> [2] Hou, Wenjun, et al. "RECAP: Towards Precise Radiology Report Generation via Dynamic Disease Progression Reasoning." *Findings of the Association for Computational Linguistics: EMNLP 2023*. 2023.
>
> [3] Zhu, Qingqing, et al. "Utilizing Longitudinal Chest X-Rays and Reports to Pre-fill Radiology Reports." *International Conference on Medical Image Computing and Computer-Assisted Intervention*. Cham: Springer Nature Switzerland, 2023.
>
> [4] Bannur, Shruthi, et al. "Learning to exploit temporal structure for biomedical vision-language processing." *Proceedings of the IEEE/CVF Conference on Computer Vision and Pattern Recognition*. 2023.
>
> [5] Demner-Fushman, Dina, et al. "Preparing a collection of radiology examinations for distribution and retrieval." *Journal of the American Medical Informatics Association* 23.2 (2016): 304-310.

---

### Official Review · Reviewer_91pY · 2024-02-27

**Confidence:** 3
**Preliminary Rating:** 4
**Final Rating:** 4

**Summary:**

This work introduces a novel VLM called PLURAL, which is pre-trained on natural and longitudinal chest X-ray data for the diff-VQA task. The model is developed using a step-by-step approach, starting with being pre-trained on natural images and texts, followed by being trained using longitudinal chest X-ray data. The longitudinal data consist of pairs of X-ray images, along with question-answer sets and radiologist’s reports that describe the changes in lung abnormalities and diseases over time.

**Strengths:**

1. This work involves using a Transformer-based network that has been pre-trained on natural images and texts and then optimizing it using longitudinal chest X-ray data. The compared methods are relatively new.
2. The experimental results include alation studies of each module, making it reasonable.

**Weaknesses:**

1. The 3.1 section says the model performs better on disease severity, please clarify it in Fig. 3 (c).
2. Lack of specific examples of non-difference VQA tasks.
3. Due to the specific nature of medical images and reports, it may be better to include physician comments.

**Detailed Comments:**

1. Lack of specific examples of non-difference VQA tasks.
2. The 3.1 section says the model performs better on disease severity, please clarify it in Fig. 3 (c).
3. Due to the specific nature of medical images and reports, it may be better to include physician comments.

**Justification Of Final Rating:**

Thanks for the author's response. The experiment results show the method performs well on the MIMIC-Diff-VQA dataset. Based on the work and the response, I think this work makes progress in the CXR image difference VQA task, and this paper deserves accepted.

**Justification Of The Preliminary Rating:**

This work involves using a Transformer-based network that has been pre-trained on natural images and texts and then optimizing it using longitudinal chest X-ray data. The compared methods are relatively new. Due to the specific nature of medical images and reports, it may be better to include physician comments.

**Questions To Address In The Rebuttal:**

1. Provide some specific examples of non-difference VQA tasks.
2. Due to the specific nature of medical images and reports, it may be better to include physician comments.

---

> ### Author Response · Authors · 2024-03-15
>
> Thank you for the considerable time and effort you have devoted to your review.
>
> ## Q1 Examples of non-difference VQA
> Thank you so much. We have attached some examples of non-difference VQA tasks in Appendix B Fig. 5, as you proposed.
>
> ## Q2 Clarification in Fig. 3 (c).
> Thanks for the comment. As you suggested, we have elaborated on the severity and location of the abnormality in the caption of Fig. 3c for further clarification.
>
> ## Q3 Physician comments
> We really appreciate your feedback. Based on your suggestions, we added some texts and figures in Supplementary Appendix A after having the radiologist’s comments on some examples.
>
>
> In this revision, we have been able to articulate more precise statements regarding disease severity and incorporate non-difference VQA examples and physician comments. We have applied all the comments you mentioned to our paper.

---

> > ### Comment · Reviewer_91pY · 2024-03-26
> >
> > Thanks for your clarification of my doubts. I have no further questions.

---

### Official Review · Reviewer_Zajj · 2024-03-06

**Confidence:** 4
**Preliminary Rating:** 5
**Recommendation:** Oral
**Final Rating:** 5

**Summary:**

This paper presents a vision language model to address the task of difference visual question answering (diff-VQA) in the context of longitudinal chest X-ray images. The authors propose a multi-step approach to pre-train the model on natural images and texts and then pre-train/fine-tune on longitudinal chest X ray images and radiological reports. Ablation studies were conducted to understand the impact of each pre-training step of the model. Results show that the proposed model outperforms some SOTA for natural language generation metrics.

**Strengths:**

- The problem of comparing longitudinal images is of high clinical interest in the medical domain.
- Pre-training on natural images and texts, then pre-training/fine-tuning on medical data to transfer knowledge.
- The paper is clear, and well organized, and the figures help to understand the proposed methodology.
- Ablation studies help better understand the impact of pretraining steps and the input structure on the model’s performance.

**Weaknesses:**

- Missing variability assessment or statistic tests to verify the significance of the gain in each metric.
- Authors do not present an error analysis part in the qualitative results to check the clinical relevance of the errors.
- Authors do not discuss the limitations of their work.

**Detailed Comments:**

In the results, consider using cross-validation or statistical tests to verify the significance of the differences, which can be sometimes very subtle, especially in the ablation studies (Tables 2 and 3). The dataset size used for the non-difference VQA is missing (results in Table 4).

**Justification Of Final Rating:**

The authors have added error analysis to better understand the model's errors and their clinical relevance. The work is well presented and it would be of good relevance for the community working on the difference VQA task.

**Justification Of The Preliminary Rating:**

This article addresses a topic of clinical interest in the medical field that is not often addressed, namely the difference between the response to visual questions for longitudinal images. The authors propose a new VLM by introducing a multi-stage pre-training process that uses longitudinal images and radiological reports to improve the model's ability and knowledge of chest X-ray images. The authors show that their model outperforms some SOTA methods on a set of natural language generation metrics while conducting ablation studies to understand the impact of each pre-training step on model performance. An error analysis step is missing to check whether the errors made by the model have a clinical pattern or not. The results do not include an analysis of variance.

**Questions To Address In The Rebuttal:**

- What are some examples of errors made by the model and do they have any clinically relevant pattern? for example, describing the same abnormality but with different words.
- Did you test base pre-trained models other than OFA (CLIP, for example)?
- Did you perform any statistical tests or variance analysis to verify the significance of the results?

---

> ### Author Response · Authors · 2024-03-15
>
> We are grateful for the time and effort you put into your detailed review. The points you highlighted have been immensely helpful in revising our paper.
>
> ## Q1 Examples of errors and clinically relevant pattern
>
> Thanks for your valuable opinion. As you suggested, we have included an error analysis in Supplementary Appendix A. As you pointed out, we found a clinically relevant case where PLURAL generated an answer for describing the change of an abnormality in time-reverse order.
>
> ## Q2 Test on the other base pre-trained model
>
> Thank you so much. We think that the OFA model is the best fit as a baseline based on the following reasons: First, it is easy to train the OFA model using multiple tasks, such as report generation and VQA. Second, the network architecture is simple, which permits us easily to add a new input branch for a past image. Third, a pre-trained model weight is available on the web, reducing the burden of making a pre-trained model by ourselves. There had been other candidates to be used as baselines [1-2] but these are limited in some of the criteria above.
>
> ## Q3 Statistical tests or variation analysis
>
> Thanks for your comment. We are also eager to test the statistical significance of our experiments and check the impact on performance improvement, such as in the ablation study. Unfortunately, the proposed methodology requires training and finetuning a model through three stages, preventing us from repeating experiments, and requiring a lot of computational resources and costs. Therefore, we regret to say that checking the statistical significance is really hard for us under the current circumstances.
>
> [1] Radford, Alec, et al. "Learning transferable visual models from natural language supervision." *International conference on machine learning*. PMLR, 2021.
>
> [2] Alayrac, Jean-Baptiste, et al. "Flamingo: a visual language model for few-shot learning." *Advances in neural information processing systems* 35 (2022): 23716-23736.

---

> > ### Comment · Reviewer_Zajj · 2024-03-27
> >
> > Thank you for your response. I have no further questions.

---

### Meta-Review · Area_Chair_wJoe · 2024-04-04

**Recommendation:** Accept (Poster)
**Confidence:** 4

**Metareview:**

The paper introduces a novel Vision-Language Model (VLM) called PLURAL for the task of difference visual question answering (diff-VQA) in longitudinal chest X-ray images. The model is developed through a multi-step approach, starting with pre-training on natural images and texts, followed by pre-training/fine-tuning on longitudinal chest X-ray images and radiological reports. Ablation studies are conducted to analyze the impact of each pre-training step on the model's performance. The proposed model demonstrates superior performance compared to some state-of-the-art (SOTA) models for natural language generation metrics.

Overall, the paper presents a promising approach for diff-VQA in longitudinal chest X-ray images, with a clear methodology and experimental setup. However, improvements in the discussion of results, including variability assessment and error analysis, would strengthen the paper's contribution to the field.

---

### Decision · Program_Chairs · 2024-04-06

Accept (Poster)